

# Genome-wide analysis of genetic diversity and artificial selection in Large White pigs in Russia

Siroj Bakoev[1,2,*], Lyubov Getmantseva[1,*], Olga Kostyunina[1], Nekruz Bakoev[1], Yuri Prytkov[1], Alexander Usatov[3] and Tatiana V. Tatarinova[4,5,6,7]

[1] Federal Research Center for Animal Husbandry named after Academy Member LK. Ernst, Dubrovitsy, Russia
[2] Centre for Strategic Planning and Management of Biomedical Health Risks, Moscow, Russia
[3] South Federal University, Rostov-on-Don, Russia
[4] Department of Biology, University of La Verne, La Verne, CA, United States of America
[5] Department of Genomics and Bioinformatics, Institute of Fundamental Biology and Biotechnology, Siberian Federal University, Krasnoyarsk, Russia
[6] Institute for Information Transmission Problems, Moscow, Russia
[7] Vavilov Institute for General Genetics, Moscow, Russia
* These authors contributed equally to this work.

## ABSTRACT

Breeding practices adopted at different farms are aimed at maximizing the profitability of pig farming. In this work, we have analyzed the genetic diversity of Large White pigs in Russia. We compared genomes of historic and modern Large White Russian breeds using 271 pig samples. We have identified 120 candidate regions associated with the differentiation of modern and historic pigs and analyzed genomic differences between the modern farms. The identified genes were associated with height, fitness, conformation, reproductive performance, and meat quality.

Corresponding author
Lyubov Getmantseva,
ilonaluba@mail.ru

## INTRODUCTION

Human consumption drives the artificial selection of farm animals. Understanding how selection creates genetic differences between populations of different farms is essential for effective livestock development.

In the last two centuries, a common strategy was to maximize pig farming profitability of highly productive commercial breeds (such as Large White, Landrace, and Duroc) with high growth rates, good feed conversion and increased lean meat yield (*Wang et al., 2018*). As a result, these breeds became popular worldwide, including in the Russian Federation (*Traspov et al., 2016c*; *Traspov et al., 2016a*; *Traspov et al., 2016b*; *Čandek Potokar et al., 2019*; *Čandek Potokar & Nieto, 2019*).

Considering that Yorkshire pigs in Northern America are direct descendants of the European Large White lineage (*Amer et al., 2020*), Large White is the most common commercial breed group. Countries that develop production usually import the breeding stock of Large White pigs since these pigs have a flexible genetic structure adapted to

selection pressure (*Getmantseva et al., 2020*). This flexibility and genetic variation of the breed make it an exciting object for scientists striving to find the genomic regions and genes responsible for the variation.

The initial livestock of Large White pigs (approximately 100 animals) was brought to the Soviet Union from England in 1923. As a result of continuous breeding efforts, a new regional population of the Large White breed was created in the USSR during the second half of the 20th century (*Traspov et al., 2016a*; *Getmantseva et al., 2020*). The fall of the Soviet Union caused another period of hardship for Russian pig farming (*Smith, 2014*). The breeding programs were nearly stopped, farming practices deteriorated, pigs were massively affected by diseases, and were culled in huge numbers. After the USSR's collapse, the Soviet livestock was almost entirely replaced by imported pigs from the leading breeding centers of Denmark, France, England, Holland, Ireland, etc. Mitochondrial DNA analysis of pigs from various European breeding centers shows significant genetic differences (*Getmantseva et al., 2020*). In this work, we compare the Large White pigs of Soviet breeding with the modern commercial pigs. We have also analyzed the DNA structure of contemporary Large White pigs within and between the breeding farms in Russia. We have identified selection signatures attributed to the socio-economic conditions and breeding centers' practices.

## MATERIALS AND METHODS

### Animals and sample collection

According to standard monitoring procedures and guidelines, the participating holdings specialists collected tissue samples, following the ethical protocols outlined in the Directive 2010/63/EU (2010). The pig ear samples (ear pluck) were obtained as a general breeding monitoring procedure or during the slaughter. The collection of ear samples is a standard practice in pig breeding (*Kunhareang, Zhou & Hickford, 2010*). Previously collected historic tissue samples of the Soviet-bred pigs were obtained from breeding farms in Russia between 2006 and 2010.

We have assembled a pool of 271 pig samples; 99 historical examples of the Large White pigs from the Soviet breeding program (LW_Old, samples collected from four breeding farms between 2006 and 2010); 106 samples of Large White pigs of modern breeding from four Russian farms (LW_New: LW_1 = 28; LW_2 = 31; LW_3 = 26; LW_4 = 21, all samples collected between 2018 and 2020). The Landrace ($L = 23$) and Duroc ($D = 43$) samples were collected between 2018 and 2020. Genomic DNA was extracted from ear samples using a DNA-Extran-2 reagent kit (OOO NPF Sintol, Russia) following the manufacturer's protocol. The quantity, quality, and integrity of DNA were assessed using a Qubit 2.0 fluorometer (Invitrogen/Life Technologies, USA) and a NanoDrop8000 spectrophotometer (Thermo Fisher Scientific, USA).

### Genotyping

The samples were genotyped using the GeneSeek® GGP Porcine HD Genomic Profiler v1 (Illumina Inc, USA), which includes 68,516 SNPs evenly distributed with an average spacing of 25 kb. Genotype quality control and data filtering were performed using PLINK

1.9, as recommended by *Marees et al. (2018)*. The total genotyping rate is 0.999307; 41,262 variants and 271 pigs passed the QC filters and were retained for further analysis.

## Data availability

The dataset can be accessed at http://www.compubioverne.group/data/PIG/.

## Population structure analysis

To study population structure, we performed a singular value decomposition (SVD) decomposition of the GRM using the SVD function in R (*Barker et al., 2001*; *Van Raden, 2008*). R package AdmixTools was used to compute various $F_2$ statistics for all pairs of populations and $F_3$ statistics outgroup statistics estimating the relative divergence time for pairs of populations, using the Duroc pigs as an outgroup (*Lazaridis et al., 2014*; *Patterson et al., 2012*). AdmixTools was also and to plot the trees (*Patterson et al., 2012*; *Liu et al., 2019*). Using the *find_graphs* routine, we have generated and evaluated admixture graphs to find the best-fitting arrangements. Although $F_{ST}$ and $F_2$ statistics also calculate genetic distance or divergence time, they may be influenced by population sizes. Statistic $F_3$(outgroup; A, B) estimates the genetic distance between the outgroup and branching point between populations A and B (*Maier & Patterson, 2020*).

To study the genetic structure, we used the VanRaden genomic relationship matrix (GRM)) (*Van Raden, 2008*). This matrix is constructed from the SNP matrix $Z$, where rows correspond to individuals and columns to markers, as $G = \frac{ZZ'}{k}$, where denominator $k$ is calculated using the allele frequencies of genotyped individuals: $k = 2\sum_i p_i(1 - p_i)$. The denominator attains maximum when all allele frequencies are equal to $\frac{1}{2}$.

We performed the SVD decomposition of GRM using the SVD function in R. Singular value decomposition (SVD) is a valuable tool for characterizing population genetic structure to detect and extract small signals even if the data is noisy (*Berrar, Dubitzky & Granzow, 2007*). Besides, a graphics package in R based on the GRM matrix is used to visualize the relationships between the studied populations of pigs. Plots of the first and second SVD components and a heat map were generated to visualize the SVD results. We used the singular value decomposition (SVD) approach (*Golub & Reinsch, 1971*) to assess the genetic structure of the studied populations of Large White pigs in Russia.

To visualize the relationship between the studied populations of pigs using the graphics package in R, based on the GRM matrix, we built a heatmap plot that has separated the pigs by breeds. (R: A Language and Environment for Statistical Computing, http://www.r-project.org).

## Detection of selection signatures

We used two statistics that can be calculated for unphased genotypic data: $F_{ST}$ and $F_{LK}$. Fixation index $F_{ST}$ is a measure of population differentiation due to genetic structure. It is frequently estimated from genetic polymorphism data, such as single-nucleotide polymorphisms (SNP) or microsatellites. $F_{ST}$ value of a locus is calculated as a ratio of the variance of allele frequencies between the populations and the sum of the variances within and between populations. Positive selection is indicated by high $F_{ST}$ values relative to their heterozygosity (*Weigand & Leese, 2018*). Smoothing of $F_{ST}$ isused to identify contiguous
genomic regions under selection. The smoothed $F_{ST}$ method is based on the pure drift model of *Nicholson et al. (2002)*. According to this model, individual SNPs are grouped into genomic windows, and their average smoothed $F_{ST}$ values are calculated. Smoothed $F_{ST}$ isuseful for analyzing distantly related populations and reveals subtle differences between them *Porto-Neto et al. (2013)*.

We compared LW_OLD and LW_New groups using the $F_{ST}$ analysis to find genomic traces of recent selection resulting from different socio-economic conditions. We compared pigs from different farms to analyze how the selection centers' preferences and breeding practices affect the genomes. Then each farm was compared to the rest of the subgroups combined. SNP regions with smoothed $F_{ST}$ values above the 95th quantile indicate positive selection; the gene content of each region was analyzed.

$F_{LK}$ is a population differentiation statistic. The calculation incorporates a kinship matrix representing the relationship between populations. $F_{LK}$-based methods are optimally effective when working with closely related populations (*Bonhomme et al., 2010*; *Fariello et al., 2014*). The $F_{LK}$ test is an extension of the Lewontin and Krakauer (LK) test (*Lewontin & Krakauer, 1973*), which takes into account both hierarchical structure between populations and population size heterogeneity by modeling the genetic discrepancy between populations resulting from population drift and division (*Bertolini et al., 2018*). We used the *hapflk* software (*Gautier, 2015*) (https://forge-dga.jouy.inra.fr/projects/hapflk). $F_{LK}$ was used to compare the LW_OLD vs. all LW_New groups. We have estimated the False Discovery Rate (FDR) for SNP identified by $F_{LK}$ and $F_{ST}$ using the *qvalue* package in R (*Storey et al., 2021*); we assumed the FDR cut-off of 0.15.

### Functional analysis

*Ensembl!* Annotation of *Sus scrofa* 11.1, https://www.ensembl.org/index.html was used to analyze genes in the identified regions. Gene set enrichment analysis with Fisher's Exact test was done using the PANTHER database (http://www.pantherdb.org/). We have also studied the GWAS literature for humans and animals for all identified genes.

## RESULTS

Admixture (*Alexander, Novembre & Lange, 2009*) analysis was conducted for $K = 2, \ldots, 20$ (Fig. 1). Across all K values, only Duroc, LW_2, and LW_4 were represented by a single admixture component. Even at $K = 2$, the population structure of the Large White pigs is visibly complex, and it can be partitioned into subpopulations that generally agree with the farm of origin. Also, the admixture profiles of modern LW pigs are different from the Soviet-bred LW pigs.

The smallest cross-validation error was obtained for $K = 10$ (Fig. S1), and we have used this value in the subsequent calculations. We used Kullback–Leibler distance to partition each population into subpopulations. Large White 2 (LW_2) was a homogeneous population, LW_3 had two subpopulations (31,17, 9); Three groups of pigs were partitioned into three subpopulations: Duroc (17, 17, 9), Landrace (20, 2, 1), and LW_4 (18, 2, 1); LW_Old were divided into four subpopulations (36, 31, 19, 13). Next, we have applied GPS (*Elhaik et al., 2014*) to test the assignment accuracy using the leave-one-out validation

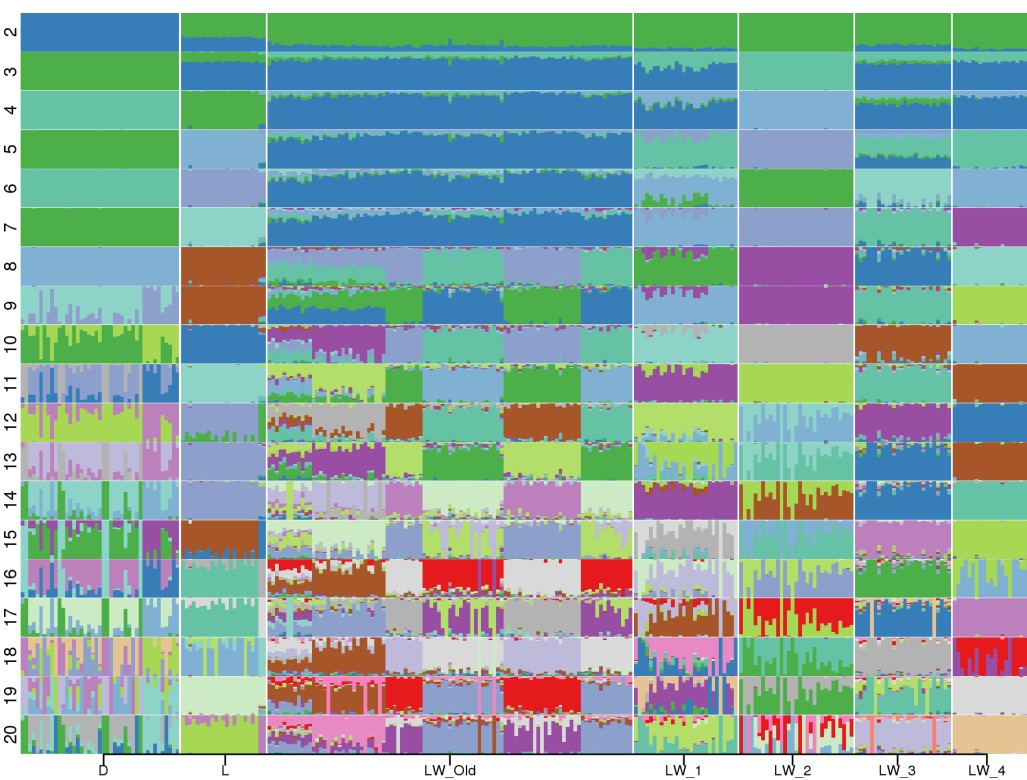

**Figure 1** Admixture profiles for $K = 2...20$. Pig breeds are designated on the horizontal axis. D, Duroc; L, Landrace; LW_Old, USSR Large White pigs; LW_1, LW_2, LW_3, LW_4—modern Large White pigs from different farms. Cross-validation analysis shows that $K = 10$ has the smallest CV error.

procedure for subgroups with at least two members: all subpopulation labels were correctly recovered. Therefore, there is a possible lack of genetic continuity between the Soviet and Russian pig breeding and the absence of common breeding standards.

To investigate this further, we computed $F_2$ statistics for all pairs of populations and $F_3$ outgroup statistics, estimating the relative divergence time for pairs of populations, using the Duroc pigs as an outgroup (Table 1). Figure 2 shows the best graph (identified by the *find_graphs* routine) connecting the studied old and new populations. $F_3$ analysis shows that although the modern Large White pigs differ from farm to farm, they share more with each other and with the old Large White pigs than with the Landrace pigs.

We used the singular value decomposition (SVD) approach (*Golub & Reinsch, 1971*) to assess the genetic structure of the studied populations of Large White pigs in Russia. Figures 3 (A, B, C) shows the SVD analysis output in axes PC1/PC2, PC1/PC3, and PC2/PC3. Pre-defined breed/farm groups correspond to well-separated clusters. In Fig. 3D (Heatmap), all Large White pigs formed a different cluster, separated from the Duroc and Landrace breeds. The same trend can be seen in the PCA plot (Fig. 3A). PCA plots show (Fig. 3C) clear separation of Durok, Landrace, and all Large White breeds. Also, LW pigs from different farms form separate clusters. This result agrees with the admixture, $F_2$, and $F_3$ analyses.

**Table 1** F3 outgroup analysis.

| A | B | f3 estimate | Standard error | z-score | p-value |
|---|---|---|---|---|---|
| LW_1 | LW_2 | 0.090 | 1.78E−03 | 50.82 | 0.00E+00 |
| LW_1 | LW_4 | 0.088 | 1.77E−03 | 49.73 | 0.00E+00 |
| LW_2 | LW_4 | 0.088 | 2.43E−03 | 36.08 | 4.35E−285 |
| LW_2 | LW_3 | 0.081 | 2.32E−03 | 35.13 | 2.22E−270 |
| LW_4 | LW_Old | 0.081 | 1.93E−03 | 41.77 | 0.00E+00 |
| LW_3 | LW_4 | 0.080 | 2.23E−03 | 36.03 | 2.76E−284 |
| LW_1 | LW_3 | 0.080 | 1.51E−03 | 52.98 | 0.00E+00 |
| LW_2 | LW_Old | 0.079 | 1.80E−03 | 43.79 | 0.00E+00 |
| LW_1 | LW_Old | 0.078 | 1.21E−03 | 64.29 | 0.00E+00 |
| LW_3 | LW_Old | 0.075 | 1.63E−03 | 45.97 | 0.00E+00 |
| L | LW_4 | 0.064 | 2.10E−03 | 30.53 | 1.10E−204 |
| L | LW_1 | 0.063 | 1.81E−03 | 35.10 | 6.74E−270 |
| L | LW_3 | 0.063 | 1.60E−03 | 39.64 | 0.00E+00 |
| L | LW_2 | 0.063 | 2.02E−03 | 30.93 | 5.06E−210 |
| L | LW_Old | 0.061 | 1.54E−03 | 40.00 | 0.00E+00 |

Next, we used the smoothed $F_{ST}$ and the $F_{LK}$ to compare the modern and historic Russian pig populations. There were 120 genes associated with the differentiation between old and new Large White breeds: 52 were identified by the $F_{ST}$ method and 68 by the $F_{LK}$method; both approaches flagged 16 genes.

$F_{ST}$ analysis has identified several genomic regions. Chr1: 51753405-51918328 (4 SNPs), Chr4: 81493481-83928709 (29 SNPs), Chr4: 127146360-128244327 (23 SNPs), Chr5: 7147061-72783062 (36 SNPs), Chr6: 45595002-45854092 (7 SNPs) and Chr6: 106647339-121553230 (95 SNPs) (Table S1). According to the $F_{LK}$ method with the most significant signals, 158 SNPs were identified (Fig. 4, Table S2), of which 27 SNPs on Chr6: 107577819-12095419 were also determined by the $F_{ST}$ method.

After determining the overlap of the identified SNPs with the known QTLs, we have found that despite producing different SNP lists, $F_{ST}$ and the $F_{LK}$ methods have resulted in nearly identical QTLs lists. According to either method, the identified areas overlap with quantitative trait loci (QTLs) for traits related to meat and anatomical characteristics, animal fitness, meat color, meat quality, conformation indicators of pigs, defects, susceptibility to diseases, blood biochemistry, reproductive traits (fertility and reproductive organs), and productivity traits (growth and development) (Table S3 and S4).

Based on the pathway enrichment analysis, nine significant pathways were identified by $F_{ST}$ (Table 2). The genes detected by both approaches belong to six pathways: Synaptic vesicle trafficking (regulates the processes of the nervous system), T-cell activation (ensures the functioning of the immune system), Alzheimer disease-amyloid secretase pathway (responsible for brain function and behavior), Muscarinic acetylcholine receptor 1 and 3 signaling pathway (participates in the peripheral nervous system, controls parasympathetic reactions), PDGF signaling pathway (responsible for the structural and functional

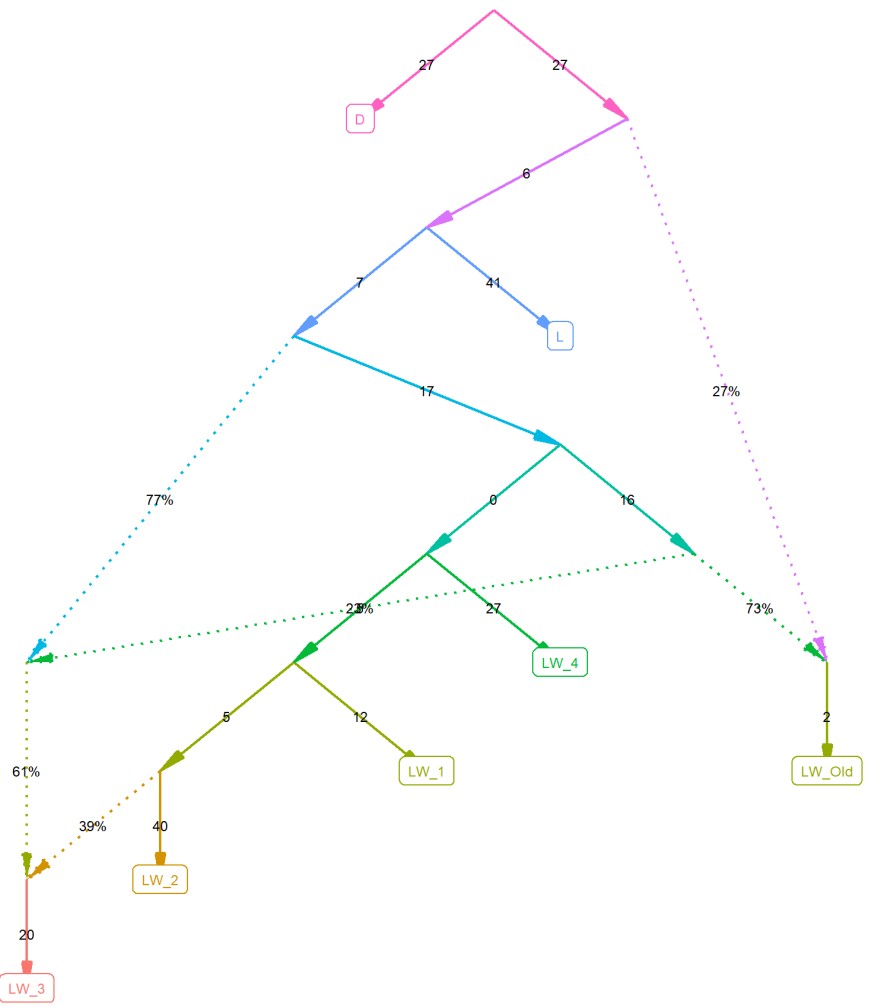

**Figure 2** **Admixture graph computed using f-statistics.** The plot is done using the ADMIXTOOLS2 R package, using the Duroc pigs as an outgroup.

development of the body), and Gonadotropin-releasing hormone receptor pathway (controls reproductive function).

Genomic signatures of different breeding practices were analyzed by comparing subgroups from the LW_New group using the smoothed $F_{ST}$. In pigs LW_1, a strong signal was detected on Chr14: 45509383-46738288 (24 SNPs) (Table S5). This region contains 21 QTLs, 17 of which are associated with reproductive traits (twelve QTLs—Number of mummified pigs, four QTLs—Litter size, and one QTL—Litter weight total) (Table S6). Also, there are three QTLs for production traits ("Ratio of lifetime non-productive days to herd life" and "Bodyweight at birth") and 1 QTL Health trait ("White blood cell number"). In this region, seven genes were identified (*AP1B1, EWSR1, KREMEN1, NEFH, THOC5, TTC28, ZNRF3*) (Table S5). The enrichment analysis has identified the Wnt signaling pathway involved in regulating embryonic development (Table 3).

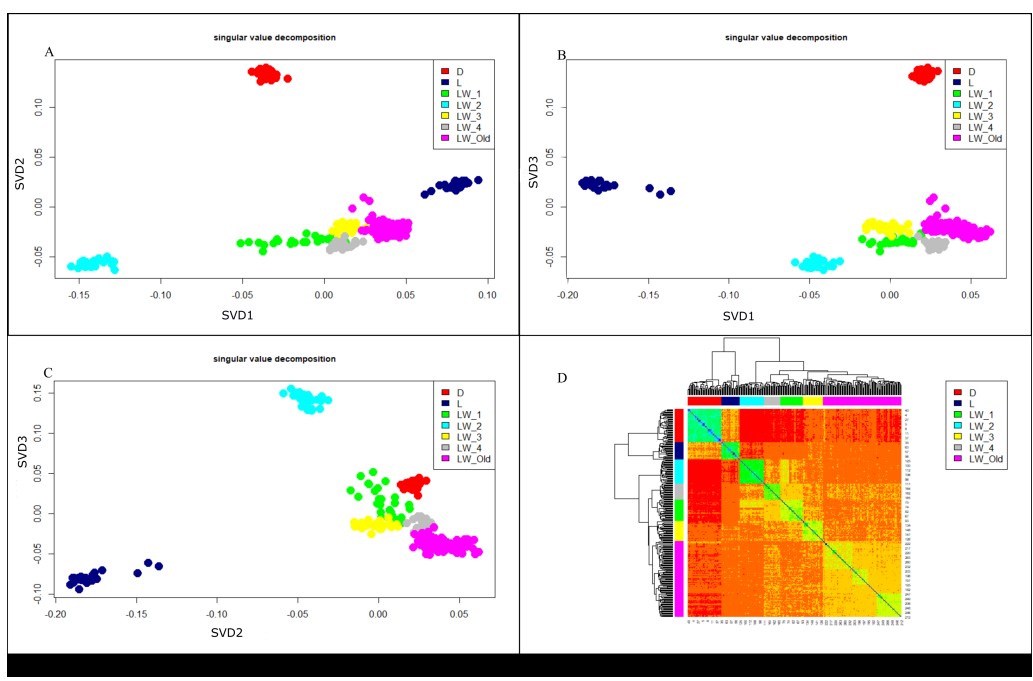

**Figure 3 SVD (A, B, C) and Heatmap (D) for pigs.** D, Duroc; L, Landrace; LW_Old - Large White Russian selection, LW-1, 2, 3, 4—a commercial Large White breed. Each LW group corresponds to a different breeding farm.

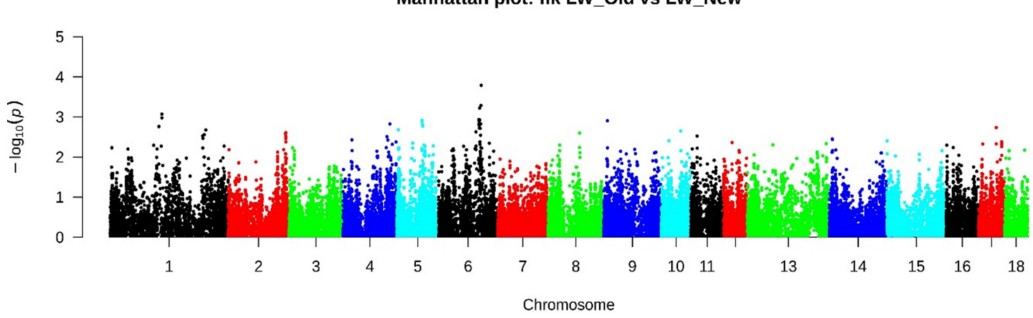

**Figure 4 Manhattan plot LW_OLD vs. LW_NEW based on $F_{LK}$ results.** According to the $F_{LK}$ method with the most significant signals, 185 SNPs were identified, of which 2 SNPs localized in chr4: 82530260-83269208 and 27 SNPs in chr6: 107577819-12095419 were also determined by the $F_{ST}$ method.

In LW_2, signals were found on Chr1: 62307146-66047984 (70 SNPs) (Table S7). This area overlaps with 5 QTLs: Reproductive traits (Litter size), Exterior traits (Behavioral and Body shape), and Meat and carcass traits (Palmitic acid content and Ham weight) (Table S8). In this region, nine genes were identified (*FBXL4, FHL5, GPR63, KLHL32, MANEA, MMS22L, POU3F2, U6, UFL1*) (Table S7). No known pathways were significantly enriched.

In LW_3, signals were found on SSC6: 130871443-133500969 (44 SNPs) and SSC15: 77070266-77566996 (Table S9). These areas overlap with 89 QTLs, of which 85 QTLs are

**Table 2** **Major gene pathways identified by the F$_{ST}$ method (*LW_Old vs. LW_New*).** Based on the pathway enrichment analysis, 37 pathways were identified.

| # | PANTHER pathways | Expected | Fold enrichment | raw *P*-value |
|---|---|---|---|---|
| 1 | Synaptic vesicle trafficking | 0.07 | 14.95 | 6.67E−02 |
| 2 | Biosynthesis of purines de novo (De novo purine biosynthesis) | 0.08 | 11.88 | 8.27E−02 |
| 3 | General transcription regulation | 0.09 | 10.78 | 9.06E−02 |
| 4 | Signaling pathway of muscarinic acetylcholine receptor 1 and 3 signaling pathway ) | 0.14 | 7.24 | 1.31E−01 |
| 5 | Transcription regulation by the bZIP transcription factor | 0.14 | 7.13 | 1.33E−01 |
| 6 | Alzheimer disease-amyloid secretase pathway | 0.15 | 6.53 | 1.44E−01 |
| 7 | T cell activation | 0.20 | 4.98 | 1.84E−01 |
| 8 | PDGF signaling pathway | 0.31 | 3.26 | 2.66E−01 |
| 9 | Path to the receptors of gonadotropin-releasing hormone (Gonadotropin-releasing hormone receptor or pathway) | 0.54 | 1.87 | 4.17E−01 |

**Table 3** **PANTHER pathways identified by the F$_{ST}$ method in pigs from the *LW_New* group.** Genomic regions under selection for each sub-group from the LW_New group were determined using the smoothed F ST method. Corresponding pathways were determined using the PAN-THER database.

| No | | PANTHER pathways | Expected | Fold enrichment | raw *P*-value |
|---|---|---|---|---|---|
| 1 | LW_1 | Wnt signaling pathway | 0.08 | 11.49 | 8.20E−02 |
| 2 | LW_2 | Unclassified | 6.16 | 1.14 | 1.00E−00 |
| 3 | LW_3 | Unclassified | 3.52 | 1.14 | 1.00E−00 |
| 4 | LW_4 | General transcription regulation | 0.03 | 28.68 | 3.52E−02 |
| 5 | LW_4 | Transcription regulation by the bZIP transcription factor | 0.05 | 19.12 | 5.19E−02 |
| 6 | LW_4 | Parkinson disease | 0.09 | 11.72 | 8.28E−02 |
| 7 | LW_4 | Cadherin signaling pathway | 0.11 | 9.16 | 1.04 E−01 |
| 8 | LW_4 | Huntington disease | 0.14 | 7.12 | 1.32 E−01 |
| 9 | LW_4 | Wnt signaling pathway | 0.24 | 4.13 | 2.17 E−01 |

**Table 4** **Main gene pathways identified by the F$_{ST}$ method in pigs from the *LW_New* group.** Based on the enrichment analysis, one pathway was identified in LW_1, LW_2, and LW_3 pigs, while six pathways were identified in LW_4 pigs.

| No | | PANTHER pathways | Expected | Fold enrichment | raw *P*-value |
|---|---|---|---|---|---|
| 1 | LW_1 | Wnt signaling pathway | 0.08 | 11.49 | 8.20E−02 |
| 2 | LW_2 | Unclassified | 6.16 | 1.14 | 1.00E−00 |
| 3 | LW_3 | Unclassified | 3.52 | 1.14 | 1.00E−00 |
| 4 | LW_4 | General transcription regulation | 0.03 | 28.68 | 3.52E−02 |
| 5 | LW_4 | Transcription regulation by the bZIP transcription factor | 0.05 | 19.12 | 5.19E−02 |
| 6 | LW_4 | Parkinson disease | 0.09 | 11.72 | 8.28E−02 |
| 7 | LW_4 | Cadherin signaling pathway | 0.11 | 9.16 | 1.04 E−01 |
| 8 | LW_4 | Huntington disease | 0.14 | 7.12 | 1.32 E−01 |
| 9 | LW_4 | Wnt signaling pathway | 0.24 | 4.13 | 2.17 E−01 |

associated with Meat and carcass traits (of which 68 QTLs are responsible for Conductivity 45 min post-mortem) (Table S12). 4 genes were identified in this region (*ADGRL2, GORASP2, METTL8, TLK1*) (Table S9). The gene *GORASP2* was also identified in strong outliers, determined by the $F_{LK}$ LW_Old vs. LW_New method. No known pathways were significantly enriched.

In LW_4, signals were found on Chr 4: 123920076-125079457 (24 SNPs), 6: 19730662-20040749 (13 SNPs) and Chr 9: 74428112-76248997 (32 SNPs) (Table S11). These areas overlap with four QTLs: Reproduction traits (Corpus luteum number and Teat number) and Meat and carcass traits (Intramuscular fat content and Conductivity 45 min post-mortem) (Table S12). In this area, 21 genes were identified, one of which *ASB4* was also identified in the area of strong outliers, based on the comparison of old and new pigs (Table S11). We have identified six enriched pathways (Table 3).

## DISCUSSION

In nature, individuals with the highest fitness tend to have more offspring, increasing favorable alleles in the population and leaving traces in genomes. These signatures of selection can be used to identify genomic regions under selection pressure (*Getmantseva et al., 2020*). The mechanisms underlying phenotypic differentiation induced by pig breeding have been investigated using genome-wide genotype data or high throuput sequencing (*Groenen et al., 2012*; *Yang et al., 2014*; *Diao et al., 2018*; *Gurgul et al., 2018*; *Xu et al., 2020b*; *Xu et al., 2020a*; *Bovo et al., 2020*). Genomic loci associated with growth traits, reproductive traits, coat color, ear shape, and other phenotypes are now known, as well as the genes that influence these traits (*Wilkinson et al., 2013*; *Zhang et al., 2018*; *Yu et al., 2020*). We compared the Large White pigs of Russian breeding (LW_Old) and modern commercial pigs (LW_New) in this work. We have identified 120 genes (52 by the $F_{ST}$ method and 68 by the $F_{LK}$ method, and 16 genes by both methods) in genomic regions associated with the differentiation of LW_OLD and LW_NEW pigs.

Gene *CNTN1* (SSC5) is a member of the neural immunoglobulin (Ig) subfamily and is involved in the formation of axonal connections in the developing nervous system (*Wang et al., 2019*). In vertebrates, the contactin family (CNTN) includes six related cell adhesion molecules participating in the nervous system formation and maintenance and in building neural circuits. *CNTN* genes are associated with an increased risk for autism (*Lin et al., 2016*). Also, genes associated with neural processes are often represented in the genomic regions associated with animals' domestication (*Alberto et al., 2018*). The B4GALT6 (SSC6) gene encodes lactosylceramide synthase, an essential enzyme for the biosynthesis of glycolipids. The GAREM1 gene (SSC6) encodes an adapter protein that functions in a signaling pathway mediated by the epidermal growth factor (EGF) receptor. B4GALT6 and GAREM1 are likely responsible for cardiac abnormalities, causing pigs to die during transportation (*Zurbrigg, 2013*; *Zurbrigg et al., 2017*).

The FHOD3 and DTNA genes are considered candidate genes associated with hypertrophic cardiomyopathy, a heart disease that affects all age groups (*Liu et al., 2017*; *Qing et al., 2017*), being the most common cause of heart failure and sudden death. FHOD3

gene plays a role in the polymerization of actin filaments in cardiomyocytes, while DTNA belongs to the dystrobrevin subfamily and the dystrophin family. Lack of dystrophin causes Duchenne muscular dystrophy and Becker muscular dystrophy (*Tsoumpra et al., 2020*). In pigs, dystrophin gene SNPs are associated with alterations in the accumulation of the dystrophin protein in skeletal muscle. It is related to sudden death caused by stress (*Joshua et al., 2015*).

The MPZL1 (SSC4) gene, also known as PZR, is a cell surface glycoprotein belonging to the immunoglobulin superfamily (*Jia et al., 2014*). In studies by *An et al. (2020)*, the MPZL1 gene was identified as a candidate gene for growth in Simmental beef cattle.

Modern pigs of the Large White breed, relative to pigs of the Soviet breeding program, are distinguished by a high growth rate, good feed conversion, and a smaller fat thickness. They are used in the breeding system as a mother breed and are highly fertile. The affected loci are responsible for productive traits (growth, feed conversion), fitness (fat content and percentage), reproductive traits (number of piglets at birth, multiple births, nest weight at birth). However, based on the results obtained, it can be noted that significant genetic differentiation between the study groups is due to changes in the loci responsible for the quality of meat. Over the past decades, the efforts of commercial pig breeders to increase production efficiency by accelerating animals' growth and development have led to a decrease in the quality indicators and technological properties of meat. These changes were mainly reflected in the content of intramuscular fat and the composition of fatty acids. These indicators determine muscle color, texture, water retention capacity, and the nutritional value of meat.

We have detected signals in areas responsible for animals' brain and nervous system functions and behavioral characteristics. We hypothesize that it may be due to the artificial selection of well-behaved, obedient animals for breeding.

The study of the modern livestock of Large White pigs, stratified by the selection centers, made it possible to identify individual characteristics in each subgroup. In LW_1, the signatures of modern artificial selection were identified in the genome regions mainly responsible for reproductive traits. QTLs for the number of mummified piglets were overrepresented in this area. We speculate that intensive breeding practices aimed to increase the saws' fertility can serve as one reason for mummified piglets since the limited volume of the uterus can lead to embryonic death during days 30 - 115 of fetal development. LW_2 shows a signal in the area associated with a complex of traits responsible for Litter Size and quality, Behavioral, Fatty acid content, Anatomy, and Conformation. LW_3 breeding efforts were focused on meat properties, such as Conductivity 45 min post-mortem, Back Fat, Ham and Loin weight, Dressing, Lean meat, and muscle protein percentage. Signatures of artificial selection in LW_4 were evident in Corpus luteum number, Teat number, Intramuscular fat content, and Conductivity 45 min post-mortem.

Since the same phenotype can result from multiple genotypes, similar breeding strategies can result in the same phenotype but different genotypes. Our results suggested that the main emphasis in selecting modern Large White pigs is aimed at productive characteristics, quality, and technological parameters of meat. This hypothesis can be further tested when a larger sample becomes available.

## CONCLUSIONS

To identify the putative areas under selection associated with prevailing trends in various socio-economic conditions and the specific practices and preferences of selection centers, we compared large white pigs of USSR selection and modern Russian commercial livestock. As a result, we found possible selection signals related to traits of height, fitness, conformation, reproductive performance, and meat quality and suggested genes that may act as candidate genes for these traits. These regions can be carefully tested using a larger set of pig samples. We have also identified possible genetic discontinuity between the Soviet-bred and modern Russian pigs.

### Funding

This research was supported by the Russian Science Foundation (RSF) Project No.19-16-00109 (Genotyping of Large White Pigs), and RSF Project No. 19-76-10012 (Genotyping of Landrace and Duroc pigs). The funders had no role in study design, data collection and analysis, decision to publish, or preparation of the manuscript.

### Grant Disclosures

The following grant information was disclosed by the authors:
Russian Science Foundation (RSF): 19-16-00109, 19-76-10012.
Russian Foundation for Basic Research: 19-016-00068.

### Competing Interests

Tatiana V. Tatarinova is an Academic Editor for PeerJ.

### Author Contributions

- Siroj Bakoev conceived and designed the experiments, analyzed the data, prepared figures and/or tables, authored or reviewed drafts of the paper, and approved the final draft.
- Lyubov Getmantseva conceived and designed the experiments, performed the experiments, analyzed the data, prepared figures and/or tables, authored or reviewed drafts of the paper, and approved the final draft.
- Olga Kostyunina, Nekruz Bakoev, Yuri Prytkov and Alexander Usatov performed the experiments, authored or reviewed drafts of the paper, and approved the final draft.
- Tatiana V. Tatarinova analyzed the data, prepared figures and/or tables, authored or reviewed drafts of the paper, and approved the final draft.

### Animal Ethics

The following information was supplied relating to ethical approvals (i.e., approving body and any reference numbers):

Breeding work with farm animals does not require the approval of the ethics committee in Russia. In addition, we did not work with animals, we have received animal samples from the breeders.

## Data Availability

The data is available at http://www.compubioverne.group/data-and-software/ under "# 4. Finding signatures of artificial selection in Large White pigs in Russia using genomic data".

## Supplemental Information

Supplemental information for this article can be found online at http://dx.doi.org/10.7717/peerj.11595#supplemental-information.

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
