# Peer review of "Genome-wide analysis of genetic diversity and artificial selection in Large White pigs in Russia"

_PeerJ, doi:10.7717/peerj.11595_

## Round 0.1 · original submission · Minor Revisions

I have now heard back from 2 referees who are both quite encouraging about your submission. While being positive overall, each also caught a few minor issues to be addressed prior to publication and both point out the same issue with the rate of false positives. Because false positive rates are high in genome scan studies, reproducibility tends to be low, leading the first referee to request that the conclusions be toned down. Without specific data to show selection, or explicit statistical tests for the likely rate of false positives in your study, some discussion of the power and caveats of your work seems appropriate to me as well. I agree with both referees that specific evaluation of the power of your tests and false positive rate would benefit the manuscript.

Other than those specific issues, I expect the minor changes suggested by the referees should be simple to address. If you decide to undertake the suggested revisions, please ensure that all review comments are addressed in a rebuttal letter that outlines point-by-point how you have addressed each comment. Any edits or clarifications mentioned in the rebuttal letter should also be inserted into the revised manuscript where appropriate. It is a common mistake to address reviewer questions in the rebuttal letter but not in the revised manuscript. If a reviewer raised a question, then your readers will probably have the same question so you should ensure that the manuscript can stand alone without the rebuttal letter. Directions on how to prepare a rebuttal letter can be found at: https://peerj.com/benefits/academic-rebuttal-letters/ if you need additional guidance.

I look forward to seeing your revised manuscript, and thank you for selecting PeerJ to publish your work.

Reviewer 1 ·

Basic reporting

This work presents a differentiation analysis between 'old' (2006-2010) and 'recent' (2018-2020) Large White pigs from the Russian federation. The goal was to find regions under potential selective forces that differ between the two groups of populations. A main issue with this approach is that results can be due to different causes other than selection, including drift, random sampling. Nevertheless, the paper contains material and results of interest, but the text must also state potential sources of flase positives and tone down the implications. As in many studies of this kind, genes that fall within the positive selection category are picked to make a story but reproducibility tend to be low.

Experimental design

A main drawback with this study is that there does not seem to be a big span between the 'old' and 'new' samples, and that differences found cannot only be assigned to changes brought about by selection.

Validity of the findings

Results should be assessed somewhat more critically.

Additional comments

Results are over interpreted and should be evaluated more critically, both in introduction as in discussion. Some comparison with previous evidence on selective footprints in the pig would help the manuscript.

Minor
- Spell out WWI
- Were 'old' samples all from the same farm?
- Van raden proposes several formulae , include formula that specifies actual computations
- Results is somewhat mixed with methods, describe f2,f3 tc family of statistics and interprettaion
- How were significant fst and Flk SNPs defined or justified?
- line 210 FLK does not estimate variability ....
- line 220ff, what is the point of describing hypothetical sweeps in each population? possibly simply sampling, how repeatable are results if say a bootstrap sampling is performed?
- Dscussion: most is overinterpreted, pertinence of lines 300-305 is not clear
- tone down the conclusions

·

Basic reporting

Overall, the manuscript is clearly well written and all data are supplied and reported. While most of the tables and figures are of high quality, I am concerned about the legibility of Figure 3. It would be beneficial to export these plots in a higher resolution as the data points blend together and the subtitles are unfortunately not legible. Figure 3D in particular stands out as one that may benefit from being shown on its own, as when it is compressed with 3 other figures it becomes quite difficult to read.
There are a few minor references that need corrections (line 56-59) as well as a placeholder “XX century” (line 70). The reference on line 518 is also incomplete.

Experimental design

Primary research question is well defined, with proper context detailing an interesting knowledge gap. The authors undertook a thorough and impressive approach to evaluate Large White pig genomic data for signatures of selection. Methods are described with enough detail to replicate all analyses.

Validity of the findings

The study I do have minor concern for the lack of a statistical test to determine the rate of false positives. I believe this would strengthen the findings further if included upon revision. Conclusions are concise, compelling, and well supported by the presented data.

Additional comments

With minor revisions, the manuscript will make a valuable contribution.

---

## Round 0.2 · accepted · Accept

I have now read through your response and revised manuscript and believe that you have addressed all the concerns of the referees in your revision. While some might argue the FDR of 0.15, it seems to me that this decision is somewhat arbitrary and reflects the tolerance of the researcher for Type I vs Type II error in the experimental design. Clearly with an FDR of 0.15 you will include more false positives than if you had selected a value of 0.05 but you are clear about it and I feel that your conclusions are appropriate to your data. Therefore, I am happy to accept your revised manuscript for publication. Thank you for selecting PeerJ as the outlet for your work.